# Tissue-specific inhibition of protein sumoylation uncovers diverse SUMO functions during *C. elegans* vulval development

**Aleksandra Fergin**[1,2], **Gabriel Boesch**[1,2], **Nadja R. Greter**[1,2], **Simon Berger**[1,2], **Alex Hajnal**[1]*

**1** Department of Molecular Life Sciences, University of Zürich, Zürich, Switzerland, **2** Molecular Life Science PhD Program, University and ETH Zürich, Zürich, Switzerland

* alex.hajnal@mls.uzh.ch

**Data Availability Statement:** All relevant data are within the manuscript and its Supporting Information files. The numerical data used to create

## Abstract

The sumoylation (SUMO) pathway is involved in a variety of processes during *C. elegans* development, such as gonadal and vulval fate specification, cell cycle progression and maintenance of chromosome structure. The ubiquitous expression and pleiotropic effects have made it difficult to dissect the tissue-specific functions of the SUMO pathway and identify its target proteins. To overcome these challenges, we have established tools to block protein sumoylation and degrade sumoylated target proteins in a tissue-specific and temporally controlled manner. We employed the auxin-inducible protein degradation system (AID) to down-regulate the SUMO E3 ligase GEI-17 or the SUMO ortholog SMO-1, either in the vulval precursor cells (VPCs) or in the gonadal anchor cell (AC). Our results indicate that the SUMO pathway acts in multiple tissues to control different aspects of vulval development, such as AC positioning, basement membrane (BM) breaching, VPC fate specification and morphogenesis. Inhibition of protein sumoylation in the VPCs resulted in abnormal toroid formation and ectopic cell fusions during vulval morphogenesis. In particular, sumoylation of the ETS transcription factor LIN-1 at K169 is necessary for the proper contraction of the ventral vulA toroids. Thus, the SUMO pathway plays several distinct roles throughout vulval development.

## Author summary

Many proteins are chemically modified after they have been synthesized. In particular, conjugation with the Small Ubiquitin-like Modifier (SUMO) regulates the functions and activities of a large number of proteins in animal and plant cells.

Here, we have used the Nematode *Caenorhabditis elegans* to study the various effects of SUMO protein modification on organ development. By applying a tissue-specific protein degradation system, we could selectively block the SUMO pathway in different tissues of the animals. We focused on the development of the egg-laying organ as a model, and

the graphs in the figures are provided as supplementary Data file (S1 Data.xlsx).

**Funding:** This work was supported by grants from the Swiss National Science Foundation (www.snf.ch) no. 310030-184792 and Swiss Cancer Research (www.swisscancer.ch) no. 4377-02-2018 to AH. The funders had no role in study design, data collection and analysis, decision to publish, or preparation of the manuscript.

**Competing interests:** The authors have declared that no competing interests exist.

found that the SUMO pathway acts in multiple tissues to regulate distinct cellular functions. Finally, we show that SUMO modification of one transcription factor, called LIN-1, is necessary for the proper morphogenesis of the organ. Our results indicate that the manifold effects of the SUMO pathway can be attributed to the combined action of a distinct number of SUMO modified proteins acting in different cell types.

## Introduction

Sumoylation is an essential post-translational protein modification found in eukaryotes [1,2]. A major player in this pathway is the Small Ubiquitin-like Modifier (SUMO), which shares large structural and functional similarities with Ubiquitin. However, unlike ubiquitination sumoylation promotes or inhibits protein interactions and changes protein conformation or localization, allowing transient binding to target proteins. Since its discovery in the late 1990s, SUMO has been shown to be involved in a wide range of essential biological processes [3–6]. Studying its diverse functions and identifying specific targets has however remained challenging due to the essential roles of the SUMO pathway for animal viability and development, the low concentration of sumoylated target proteins, the constant activity of deSUMOylating enzymes and the often subtle effects caused by the modification itself [3,6]. Developing tools, which allow spatial and temporal inhibition of protein sumoylation, is therefore crucial to gain a better understanding of this protein modification.

Protein sumoylation in *C. elegans* occurs essentially in the same fashion as in higher organisms. However, contrary to mammals and other vertebrates, only one SUMO orthologue, called SMO-1, exists in *C. elegans*. This renders the worm an ideal model to study this post-translational protein modification system. Activated SMO-1 is transferred to the E2 enzyme UBC-9 by the E1 enzyme formed by UBA-2 and AOS-1, and usually attached to the substrate by a SUMO E3 ligase, such as the PIAS domain protein GEI-17 [7–9]. Though, some sumoylation reactions do not involve an E3 ligase [10]. Deconjugation of SUMO from its targets is regulated by one of four SUMO proteases, ULP-1, ULP-2, ULP-4 and ULP-5 [9].

Sumoylation is essential for *C. elegans* viability and involved in a wide range of biological processes. Especially, normal vulval fate specification has previously been shown to depend on sumoylation, as *smo-1(lf)* mutants exhibit multivulva (Muv) as well as protruding vulva (Pvl) phenotypes [11]. Vulval development is an excellent model to dissect pleiotropic phenotypes at cellular resolution and uncover the actions of genetically redundant pathways [12,13]. We therefore chose this well-established model to further dissect the different roles of the SUMO pathway during organogenesis. The vulva is formed by three out of six equivalent vulval precursor cells (the VPCs P3.p trough P8.p), which adopt one of three possible cell fates. The 1˚ fate is induced in P6.p by an epidermal growth factor (EGF) signal, termed LIN-3, which is secreted by the gonadal anchor cell (AC). A lateral signal from P6.p then activates the LIN-12 Notch signaling pathway in the neighboring VPCs P5.p and P7.p to induce the alternate, secondary (2˚) fate [14–17]. After vulval fate specification, the 1˚ VPC undergoes three rounds of cell divisions producing 8 daughter cells, while the 2˚ fated VPCs each generate 7 daughter cells in an asymmetric lineage, together forming the vulva consisting of 22 cells. The remaining distal VPCs (P3.p, P4.p and P8.p) adopt the uninduced 3˚ fate, which is to divide once and fuse with the surrounding epidermis hyp7. While the VPCs proliferate, the AC breaches two basement membranes (BMs) separating the uterus from the epidermis and invades the underlying vulval epithelium [18]. During the subsequent phase of vulval morphogenesis, the vulval cells invaginate to generate a lumen, extend circumferential protrusions and fuse with their

contralateral partner cells to form a tubular organ consisting of a stack of seven epithelial rings called toroids [12,13].

Here, we have employed a tissue-specific version of the auxin-inducible protein degradation system (AID) to inhibit the SUMO pathway either in the AC or the vulval cells [19]. This approach allowed us to determine, in which tissues protein sumoylation is necessary for normal vulval development, as well as to characterize the diverse phenotypes caused by selectively blocking the SUMO pathway. Moreover, we hypothesized that by attaching an AID tag to SUMO, we could induce the auxin-dependent degradation of sumoylated target proteins in a tissue-specific manner. To test if this approach allows the tissue-specific degradation of SUMO modified proteins, we chose LIN-1, as it has previously been shown by in vitro experiments to be sumoylated at K10 and K169 [20,21]. The ETS family transcription factor LIN-1 is essential for different aspects of vulval development. During fate specification, LIN-1 inhibits VPC differentiation by recruiting transcriptional repressors in a sumoylation-dependent and independent manner thereby repressing 1° fate-specific target genes [20,22]. During vulval morphogenesis, LIN-1 promotes the actomyosin-mediated contraction of ventral toroids [23].

We mutated the two known sumoylation sites in LIN-1 (K10 and K169), measured LIN-1 expression levels after VPC-specific degradation of AID-tagged SMO-1 and observed the vulval phenotypes caused by mutation of the SUMO sites. Our results suggest that sumoylation of LIN-1 in the VPCs at K169 is required for the proper contraction of the ventral vulA toroid ring during morphogenesis. The additional phenotypes that were only observed after degradation of GEI-17 or SMO-1 indicate that LIN-1 is one of several relevant SUMO targets during vulval development.

## Results

### Tissue-specific, auxin-inducible degradation of SUMO pathway components

To dissect the interactions of different SUMO pathway components with their substrates and identify specific targets during vulval development, we adapted the tissue-specific auxin-inducible degradation system [19]. Here, we generated tissue-specific degradation drivers expressing the TIR-1 ubiquitin ligase in four different cell types; *hlh-2p>tir-1* in the AC and VU cells before and after AC specification [24], *cdh-3p>tir-1* in the AC post specification [18], *egl-17p>tir-1* in the 1° VPC and its descendants [25] and *bar-1p>tir-1* in all VPCs and their descendants [26]. We also used an existing driver, in which TIR-1 is ubiquitously expressed under the *eft-3p>tir-1* promoter [19].

Specificity of the degradation drivers was assessed with two assays. First, we used an SL2 trans-splicing acceptor to express the mCherry fluorophore under the same promoter/enhancer as TIR-1. In this way tissue-specificity could be monitored by observing mCherry expression (**Fig 1A**). Second, we assessed the loss of target protein expression by degrading a GFP- and AID- double tagged variant of GEI-17 (GFP::AID::GEI-17) [9]. A strong decrease in GFP::AID::GEI-17 expression upon auxin treatment was only observed in tissues expressing TIR-1 (**Fig 1B**).

In addition to the existing GFP::AID::GEI-17 strain, we created the AID::SMO-1 allele *smo-1(zh140)* carrying N-terminal Degron and FLAG tags (see materials and methods). This allele can be used to inhibit any protein sumoylation that does not depend on GEI-17 or another E3 ligase and at the same time attach the AID tag to SUMO target proteins. Note that homozygous *smo-1(zh140)* animals showed a wild-type vulval morphology, but were sterile as adults. Possibly, the N-terminal AID tag interferes with a specific SMO-1 function in the germline.

We then investigated the efficiency of AID-mediated degradation of the SUMO pathway by Western blot analysis. In GFP::AID::GEI-17 animals expressing the global *eft-3p>tir-1* driver

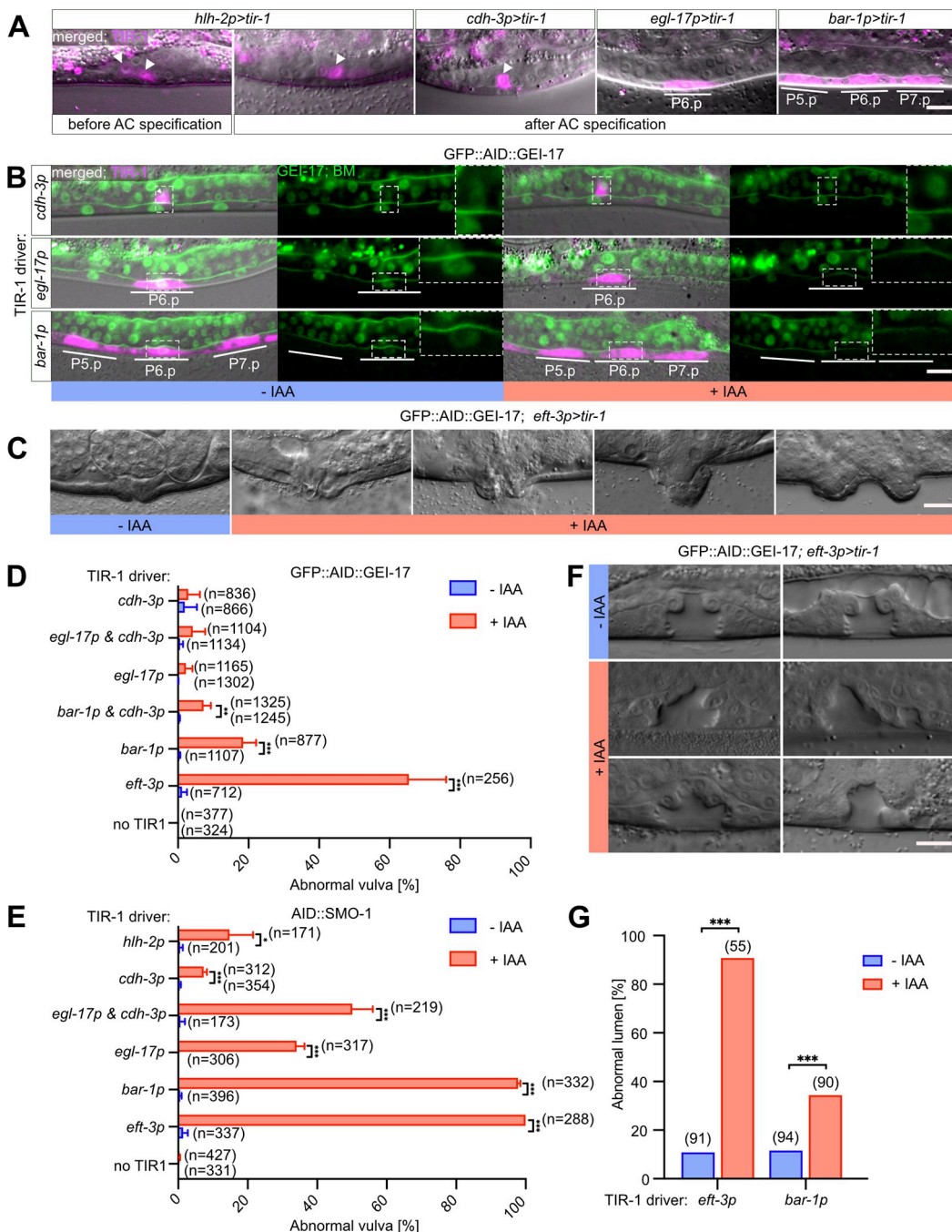

**Fig 1. Degradation of SUMO pathway components leads to abnormal vulval development.** (**A**) Tissue-specific expression of the TIR-1::SL2::mCherry degradation driver using the indicated TIR-1 drivers. mCherry expression from the bi-cistronic mRNA in magenta overlaid with the corresponding DIC images is shown for each transgene. White arrowheads indicate the AC and bars outline the location of the VPCs. (**B**) Tissue-specific degradation of GFP::AID::GEI-17 after auxin treatment using the indicated TIR-1 drivers. Left panels show the TIR-1::SL2::mCherry expression in magenta overlaid with the GFP::AID::GEI-17 and LAM-1::GFP BM markers in green and the corresponding DIC images. Right panels show only the GFP::AID::GEI-17 signal along with the LAM-1::GFP marker in green. White lines outline the location of the VPCs. The insets in the GFP panels show the region around the AC or P6.p magnified around 2.5-fold. (**C**) DIC images illustrating the vulval morphology defects in adults after global degradation of GFP::AID::GEI-17. (**D**) Penetrance of the vulval morphology defects after auxin-induced degradation of GFP::AID::GEI-17 and (**E**) AID::SMO-1 using the indicated TIR-1 drivers. The mean values ± SD obtained from three biological replicates are shown. (**F**) DIC images of L4 larvae showing an abnormally shaped vulval lumen resulting after global GFP::AID::GEI-17 degradation. (**G**) Penetrance of the vulval morphogenesis defects shown in (**F**) using the global *eft-3p* and VPC-specific *bar-1p>tir-1* drivers.

Treatment conditions are indicated as +IAA (blue) for animals treated with 1 mM auxin, and–IAA (red) for control animals. All GFP::AID::GEI-17 animals were treated at 25˚C from the L1 stage onward. All AID::SMO-1 were treated at 20˚C from the L2 to L4 stage. In **(D)** and **(E)** the numbers of animals scored are indicated in brackets. Statistical significance was determined by two-tailed unpaired t-tests **(D, E)** or with Mann-Whitney tests **(G)**. Asterisks indicate the p-values as * $p \leq 0.05$; ** $p \leq 0.01$; *** $p \leq 0.001$. The scale bars are 10 μm.

and treated with auxin, we detected a 83% (SD ± 4) reduction in GEI-17 protein levels (**Fig 2A** and **2B**). Degradation of GEI-17 did not affect SMO-1expression levels (**Fig 2A and 2C**), suggesting that depletion of GEI-17 does not significantly alter the pool of free SMO-1. By contrast, auxin treatment of AID::SMO-1 animals expressing the *eft-3p>tir-1* driver caused a 95% (SD ± 4) reduction in free AID::SMO-1 levels (**Fig 2D** and **2E**).

Taken together, the microscopic and biochemical analysis indicated that the AID system can be used to specifically and efficiently inhibit protein sumoylation in different tissues of *C. elegans*. Since global AID resulted in an almost complete depletion of the free AID::SMO-1 pool, it seems likely that this approach strongly inhibits the sumoylation of target proteins.

## Inhibiting the SUMO pathway in the VPCs or AC causes abnormal vulval development

Following the initial validation of our approach, we assessed how degradation of SUMO pathway components affects vulval development. For this purpose, we crossed the different TIR-1 degradation drivers with the GFP::AID::GEI-17 strain *gei-17(fgp1)* [27] and the N-terminally tagged AID::SMO-1 allele *smo-1(zh140)* (this study). Degradation of either protein using the global TIR-1 driver resulted in characteristic vulval morphogenesis defects (shown for GFP::AID::GEI-17 in **Fig 1C**), similar to chromosomal mutations in SUMO pathway genes [11]. Most adult animals showed a protruding vulva (Pvl) or abnormal eversion (Evl) phenotype [28] of varying severity and penetrance (**Fig 1C–1E**).

Degrading AID::SMO-1 globally using the *eft-3p>tir-1* driver resulted in almost completely penetrant vulval defects, while ubiquitous GEI-17 degradation caused abnormal vulval development in 65.5% (SD ± 10.5) of the animals (**Fig 1D** and **1E**). Global degradation caused generally more penetrant defects than tissue-specific degradation, except for depletion of AID::SMO-1 with the VPC-specific (*bar-1p>tir-1*) driver, which resulted in almost fully penetrant vulval defects (97.9% SD ± 0.6) (**Fig 1E**). The combination of the AC (*cdh-3p>tir-1*) and 1˚ VPC (*egl-17p>tir-1*) -specific drivers resulted in an additive effect (50.2% SD ± 5.8 combined versus 7.4% SD ± 0. 9 and 34.1% SD ± 2.2 separate, respectively), suggesting that the observed morphogenesis defects are a combination of separate functions played by SMO-1 in those two tissues. By contrast, degradation of GFP::AID::GEI-17 with the VPC-specific driver resulted in less penetrant defects (18.5% SD ± 3.6), and degradation using the AC and 1˚ VPC drivers alone or in combination did not result in significant defects (**Fig 1D**).

We further investigated the defects caused by GEI-17 depletion on vulval lumen formation in L4 larvae, after the toroids have been formed and the connection between vulva and uterus established (**Fig 1F** and **1G**). After global GEI-17 degradation with *eft-3p>tir-1*, 90% animals exhibited a misshaped vulval lumen, possibly due to defects in toroid fusion, cell migration defects or a failure to connect the vulva to the uterus. VPC-specific degradation using *bar-1p>tir-1*, on the other hand, had a less pronounced effects with only 34% of the animals showing abnormal vulval morphogenesis (**Fig 1F** and **1G**), suggesting that the SUMO pathway is not only necessary in the VPCs but also in other tissues.

The overall lower penetrance of vulval defects observed after degrading GFP::AID::GEI-17 compared to AID::SMO-1 may be explained by the facts that SMO-1 is the only known SUMO

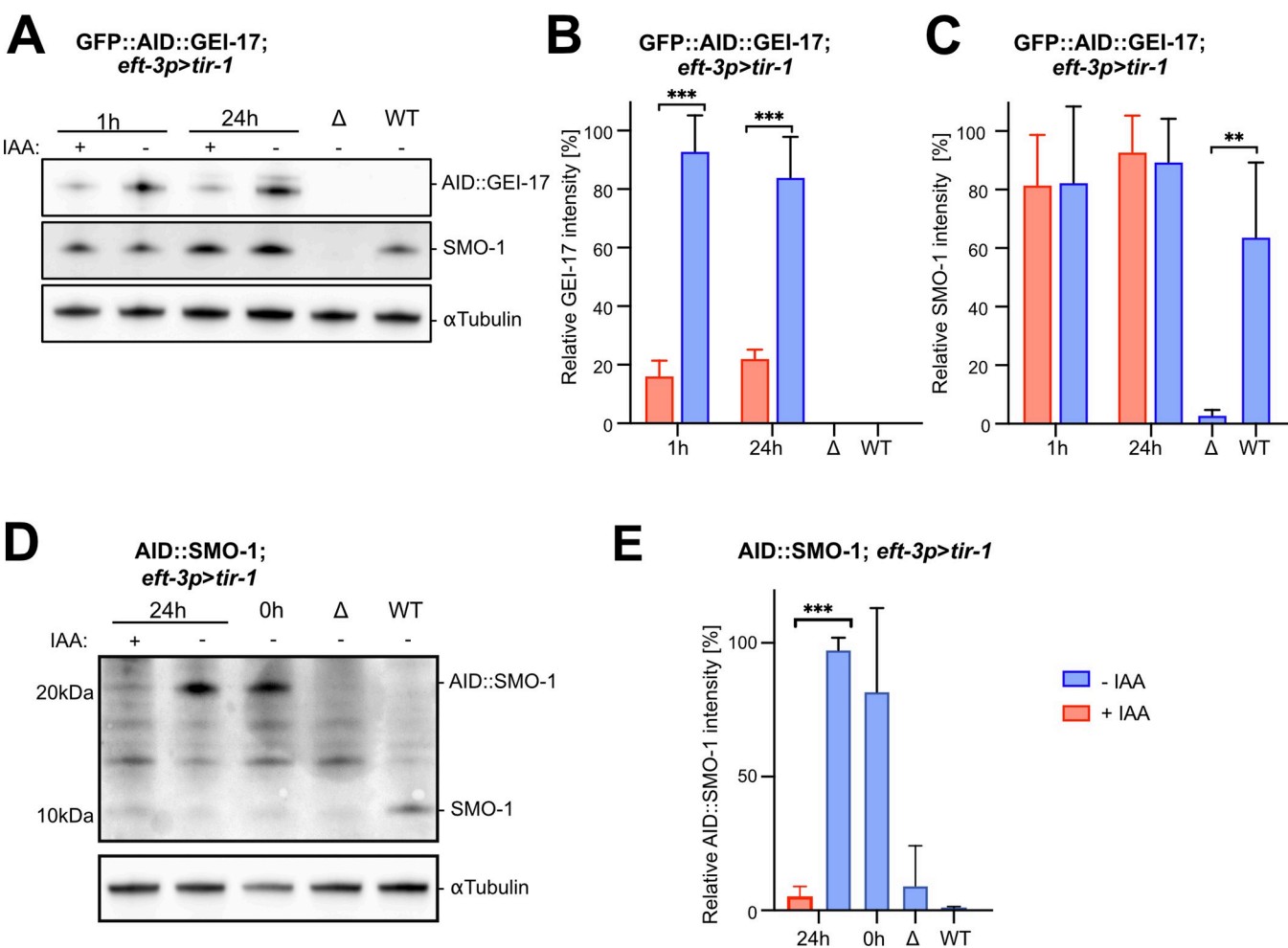

**Fig 2. Degradation of AID-tagged GEI-17 and SMO-1. (A)** Western blot analysis of GFP::AID::GEI-17 and free SMO-1 levels in one day-old adult animals carrying the *eft-3p>tir-1* driver after 1 h and 24 h (from L4 until adulthood) auxin treatment (+IAA) and untreated controls (-IAA). The GFP::AID::GEI-17 protein was detected with an anti-FLAG antibody directed against a FLAG epitope co-inserted in the *gei-17* locus. The SMO-1 protein was detected using an anti-SMO-1 antibody. Wild-type (WT) and a *smo-1(ok359)* deletion mutants (Δ) were included as controls. **(B)** Quantification of the GFP::AID::GEI-17 and **(C)** free SMO-1 protein levels in GFP::AID::GEI-17; *eft-3p>tir-1* adults after 1 h and 24 h of auxin treatment and in untreated controls. **(D)** Western blot analysis of free AID::SMO-1 levels in one-day-old adult animals carrying the *eft-3p>tir-1* driver after 24 h (from L4 until adulthood) auxin treatment (+IAA) and untreated controls (-IAA). AID::SMO-1 and untagged SMO-1 were detected using an anti-SMO-1 antibody. WT and SMO-1 deletion mutants (Δ) were used as a controls. Note the untagged SMO-1 protein in the WT controls migrating around 10 kDa. **(E)** Quantification of the AID::SMO-1 protein levels in the strains shown in **(D)**. All graphs show the mean values ± SD obtained from three independent biological replicates. Signal intensities were first normalized to the α-tubulin signals as loading control and then to the highest value in each experiment. Untreated controls are labelled with–IAA (blue) and animals treated with 1 mM auxin with +IAA (red). Statistical significance was determined by Ordinary one-way ANOVA. p-values are indicted as $^*$ p ≤ 0.05; $^{**}$ p ≤ 0.01; $^{***}$ p ≤ 0.001.

orthologue in *C. elegans*, whereas GEI-17 is not the only E3 ligase, and that not all sumoylation reactions require an E3 ligase [3,6].

## The SUMO pathway acts during all stages of vulval development

To determine the developmental stage, at which sumoylation is required for proper vulval development, we degraded SMO-1 by exposing animals to auxin at varying developmental time points between the L1/2 and L3/4 molts (**S1A Fig** and **S4 Table**), or by withdrawing auxin at different time points (**S1B Fig** and **S5 Table**) and assessing the penetrance of the observed vulval defects.

In case of the *eft-3p>tir-1* and *bar-1p>tir-1* drivers, both an early auxin treatment during L1 until the L2 molt or a late treatment beginning in L3 caused highly penetrant vulval defects. Even though there may be a slight delay until the auxin-induced effect fades after removing the animals from auxin-containing medium, these data point to a continuous action of the SUMO pathway throughout vulval development, from VPC fate specification until lumen morphogenesis.

## The SUMO pathway regulates VPC fate specification

VPC fate specification occurs between the late L2 and early L3 stages and requires the combined action of the Delta/Notch and EGFR/RAS/MAPK signaling pathways [16]. We first examined how inhibition of the SUMO pathway through VPC-specific degradation of GEI-17 affects 1° VPC fate specification. The *egl-17* gene, which encodes an FGF-like growth factor, can serve as a specific marker for the 1° VPC fate induced in reponse to EGFR/RAS/MAPK signaling [25]. We thus analyzed the expression of a transcriptional *egl-17>yfp* reporter after auxin-induced degradation of AID::GEI-17 with the *bar-1p>tir-1* driver. *egl-17>yfp* expression was stongly reduced in the 1° VPC P6.p and in its descendants at the two- (Pn.px) and four-cell (Pn.pxx) stages (**Fig 3A** and **3B**). The SUMO pathway therefore positively regulates 1° VPC fate specification.

Poulin et al. [29] and Broday et al. [11] previously reported that a loss of protein sumoylation in *smo-1(lf)* mutants or by *smo-1* RNAi caused the ectopic induction of additional VPCs besides the three proximal VPCs (P5.p to P7.p), leading to a multivulva (Muv) phenotype. To further quantify VPC fate specification after degradation of the SUMO pathway components, we counted the numbers of induced VPCs per animal after VPC-specific degradation of AID::GEI-17 (*zh142*, a *gei-17* allele containing an AID but no GFP tag) or AID::SMO-1 using the *bar-1p>tir-1* driver. Vulval induction after auxin-induced depletion of GEI-17 was slightly decreased (6.7% Vul, 2.96 VPCs/animal induced), consistent with the reduced levels of *egl-17>yfp* (**Fig 3A–3C**). Degradation of AID::SMO-1, on the other hand, resulted in a mixed phenotype with 14% of the animals showing ectopic induction and 3.5% an underinduced phenotype, but overall only a very slightly hyper-induced phenotype (3.04 VPCs/animal induced). Interestingly, we only observed ectopic induction of the posterior VPC P8.p, but never of the two anterior VPCs P3.p, P4.p (**Fig 3C**). The absence of ectopic vulval induction after GEI-17 degradation could be due to the activity of another E3 ligase acting during vulval fate specification.

Together, these data indicated that protein sumoylation in the VPCs both promotes the induction of the three proximal VPCs and inhibits the differentiation of the posterior VPC P8.p.

## Sumoylation is required for proper AC positioning and symmetrical BM breaching

Next, we analyzed the effects of the SUMO pathway by examining AC positioning as well as BM breaching. The AC in animals ubiquitously depleted of AID::GFP::GEI-17 often failed to invade at the vulval midline and sometimes did not breach the BMs or breached them in an asymmetric fashion (**Fig 3D**). In addition, the AC did not fuse in 69% of the animals to form the uterine seam cell syncytium (utse), which connects the vulva to the uterus, (**Figs 3D** and **S2D**). In many cases, the AC was not properly positioned at the vulval midline (quantified in **Fig 3E** as the angle of deflection from the midline), which may have led to the asymmetric or absent BM breaching (quantified in **Fig 3G**). Mispositioning of the AC and BM breaching defects were only observed after global, but not after VPC- or AC-specific AID::GFP::GEI-17

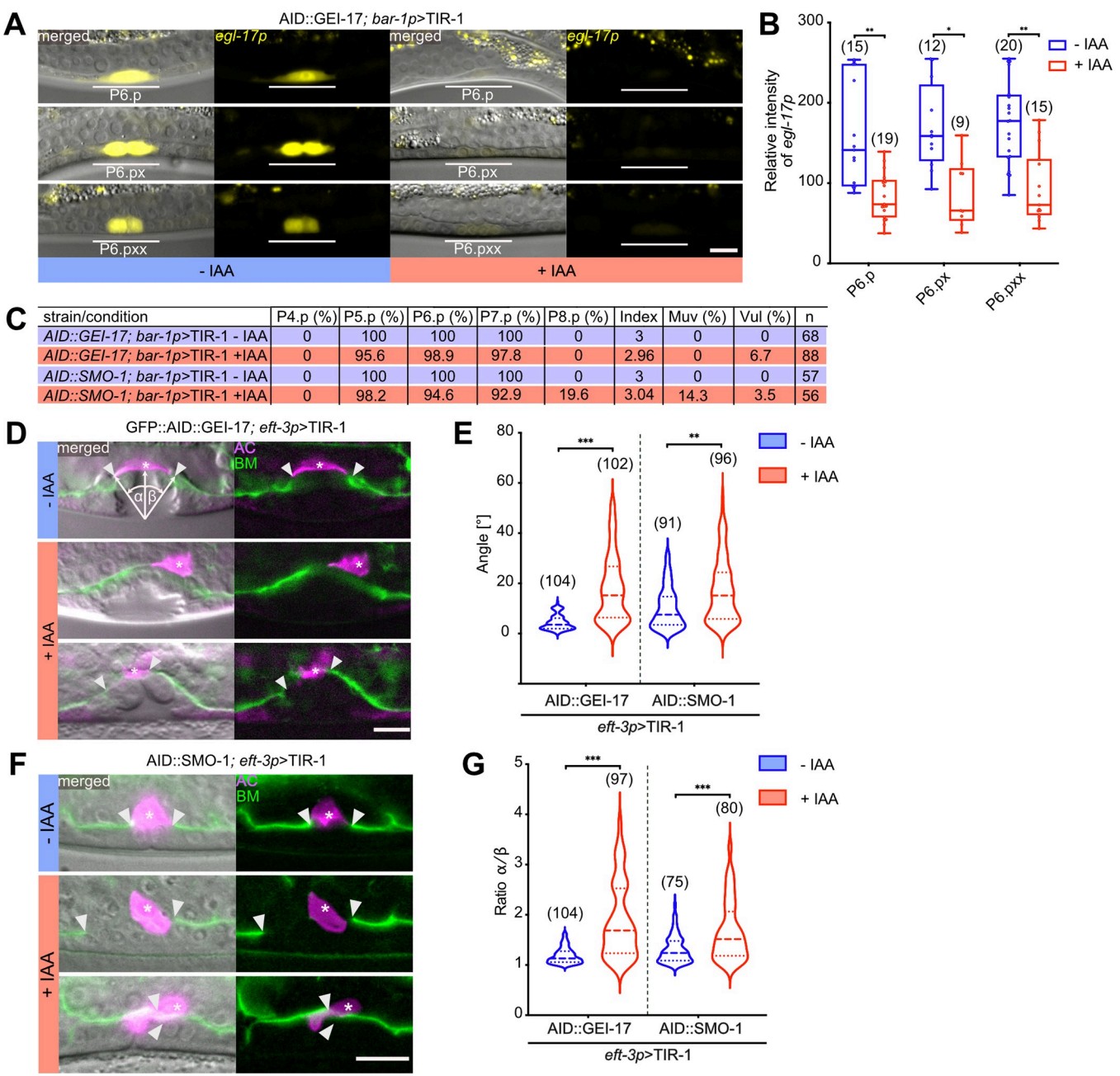

**Fig 3. The SUMO pathway regulates different stages of vulval development. (A)** *egl-17p>yfp* expression in P6.p and its descendants P6.px and P6.pxx after VPC-specific degradation of GEI-17. **(B)** Quantification of *egl-17p>yfp* expression levels after AID::GEI-17 depletion. Box plots show the median values with the 25th and 75th percentiles and whiskers indicate the maximum and minimum values. **(C)** VPC induction upon degradation of AID::GEI-17 and AID::SMO-1. For each strain and condition, the percent of induced VPCs, the average number of induced VPCs per animal (index), percent of multivulva (Muv, index>3) and vulvaless (Vul, index <3) animals, and the number of animals scored (n) are shown. **(D)** AC displacement, AC fusion defects and asymmetric BM breaching after global AID::GEI-17 and **(F)** AID::SMO-1 degradation. The BMs are labelled with LAM-1::GFP in green and the AC with *cdh-3p>*mCherry::moeABD in magenta. White arrowheads indicate the borders of the BM breaches and asterisks the AC. The left panels show the fluorescent signals merged with the corresponding DIC images. The angles α and β used to quantify AC alignment and symmetry of the BM breaching are illustrated in the top left panel. **(E)** Quantification of the AC displacement and **(G)** BM breaching asymmetry after degradation of GEI-17 and SMO-1 using the global *eft-3p>tir-1* driver. See also **S3 Fig** for the results obtained with tissue-specific *tir-1* drivers. Dashed lines in the violin plots **(E, G)** show the median values and the dotted lines the 25th and 75th percentiles. In all experiments, untreated controls are labelled with–IAA (blue) and animals treated with 1 mM auxin +IAA (red). In each graph, the numbers of animals scored are indicated in brackets. Statistical significance was determined with a Kolmogorov-Smirnov test **(B, E, G)**. p-values are indicated as * p ≤ 0.05; ** p ≤ 0.01; *** p ≤ 0.001. The scale bars are 10 µm.

degradation (**S2A** and **S2B Fig**), suggesting that signals from additional tissues besides the VPCs control AC positioning [30]. Ubiquitous degradation of SMO-1 also caused AC mispositioning and asymmetric BM breaching (**Fig 3E–3G**). As for GEI-17, neither VPC- nor AC-specific degradation of SMO-1 resulted in AC positioning or BM breaching defects (**S2A** and **S2B**)

In summary, our data indicate that the SUMO pathway is necessary for proper AC positioning and symmetrical BM breaching during invasion. This function appears to involve a cell non-autonomous activity of the SUMO pathway in tissues other than the AC and VPCs.

## The SUMO pathway is required for proper toroid morphogenesis

Since virtually all SMO-1 and most GEI-17-depleted animals showed abnormal vulval development as adults (**Fig 1D** and **1E**), while the VPC fate specification defects in L3 larvae were comparably rare (**Fig 3C**), we speculated that the inhibition of protein sumoylation perturbs normal development predominantly during the later stages of vulval morphogenesis. To characterize vulval morphogenesis in more detail, we examined the structure of the vulval toroids. To monitor toroid formation, we used either the AJM-1::GFP or the HMR-1::GFP reporter, which both label the adherens junctions between the vulval cells [31,32]. Degradation of SMO-1 or GEI-17 in the VPCs using the *bar-1p>tir-1* driver lead to a number of different defects in toroid morphology. Specifically, we observed an abnormal shape of the ventral vulA toroids (**Fig 4A**) and ectopic fusion between the vulC and vulD or the vulA and vulB1 toroids (arrows in **Fig 4A** and **4C**), similar to the defects observed in *smo-1* null mutants [11]. During vulval morphogenesis, the ventral toroids formed by the 2˚ VPCs contract in order to extend the apical lumen dorsally [23]. To quantify ventral toroid contraction, we measured the ratio of the vulA to vulB1 diameters (**Fig 4B**). The increase in the vulA/vulB1 ratio indicated that the vulA toroids did not fully contract after inhibition of the SUMO pathway in the VPCs.

Taken together, these data indicated that the SUMO pathway acts in the VPCs to control vulval toroid morphogenesis.

## AID-mediated SUMO degradation reduces LIN-1 protein levels in the VPCs

The ETS family transcription factor LIN-1 is necessary to inhibit VPC fate specification during vulval induction and for the contraction of the ventral toroids during vulval morphogenesis [20,22,23]. To investigate the role of LIN-1 sumoylation in vivo, we generated point mutations in the endogenous *lin-1* locus by replacing the two lysine residues K10 and K169 in the SUMO consensus motifs with alanine residues [20,21]. To monitor effects on LIN-1 expression levels, the two SUMO site mutations were introduced into the *lin-1(st12212)* background, in which a *gfp* tag had been inserted at the *lin-1* C-terminus. The wild-type *lin-1(st12212)* as well as the *lin-1(zh159)* K10A, K169A double mutant reporters were then crossed with the AID::SMO-1 allele together with the VPC-specific *bar-1p>tir-1* driver. Wild-type LIN-1::GFP protein expression levels decreased in the 1˚ VPC descendants once AID::SMO-1 was degraded through addition of auxin (**Figs 5A and 5B** and **S3A** and **S3B**). LIN-1::GFP expression levels in the 2˚ VPC descendants of untreated animals were lower than in the 1˚ cells, and expression in the 2˚ cells decreased only slightly after AID::SMO-1 degradation. Interestingly, the expression levels of double mutant LIN-1(K10A, K169A)::GFP in untreated animals were already lower than wild-type LIN-1::GFP, suggesting that sumoylation stabilizes LIN-1. Auxin-induced AID::SMO-1degradation did not cause a further decrease in LIN-1(K10A, K169A)::GFP levels in the 1˚ cells, indicating that the SUMO site mutations render LIN-1::GFP resistant to AID::SMO-1-mediated degradation. In the 2˚ VPC descendants, however, a slight decrease

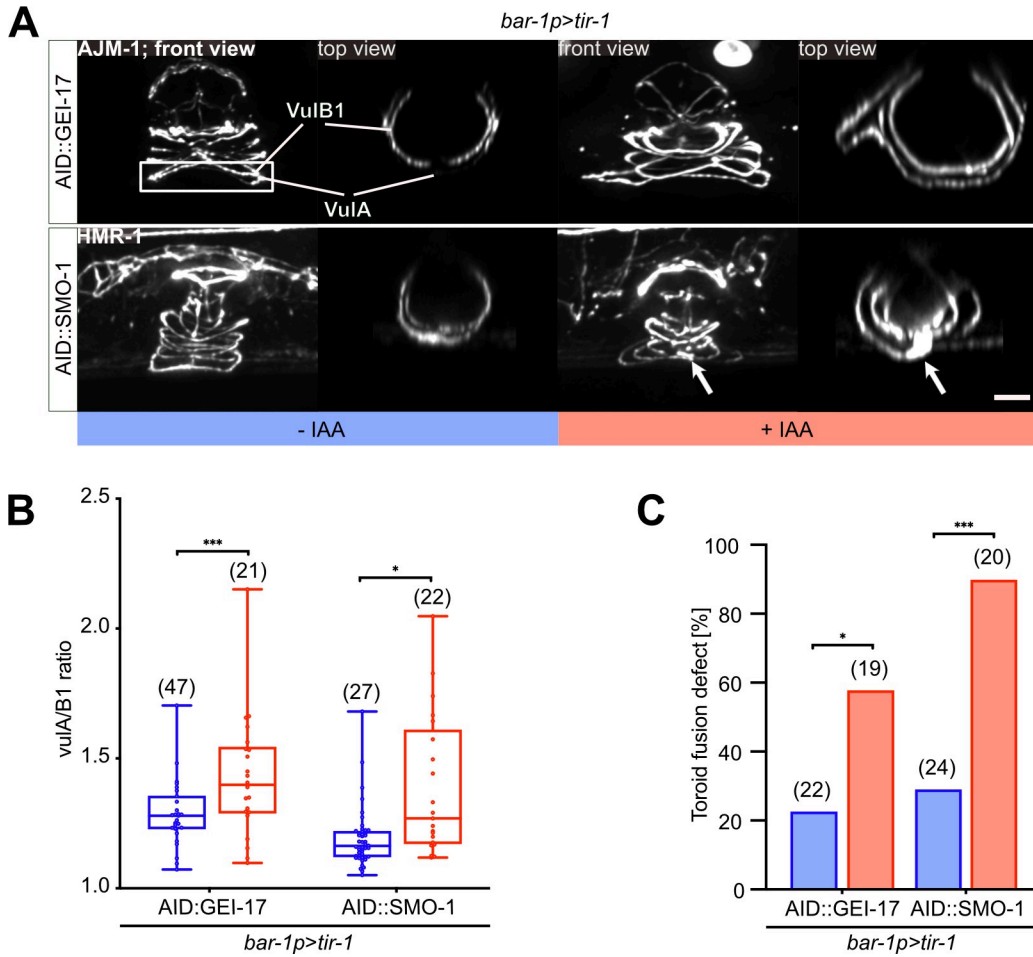

**Fig 4. Inhibition of the SUMO pathway in the VPCs causes toroid morphogenesis defects. (A)** Toroid morphogenesis defects in L4 hermaphrodites. 3D reconstructions of the adherens junctions labelled with AJM-1::GFP (for AID::GEI-17) or HMR-1::GFP (for AID::SMO-1) after VPC-specific degradation. Left panels show lateral views of z-projections. vulA and vulB1 toroids are outlined by the white rectangle in the top left panel and shown in top (xz) views in the right panels. White arrows point to abnormal fusion between the vulA and vulB1 toroids after AID::SMO-1 degradation. **(B)** Quantification of vulA contraction, calculated as the ratio of the vulA and vulB1 toroid diameter after VPC-specific AID::GEI-17 or AID::SMO-1 degradation. The box plots show the median values with the 25th and 75th percentiles and the whiskers indicate the maximum and minimum values. **(C)** Penetrance of toroid fusion defects after VPC-specific AID::GEI-17 or AID::SMO-1 degradation. In all experiments, untreated controls are labelled with–IAA (blue) and animals treated with 1 mM auxin +IAA (red). In each graph, the numbers of animals scored are indicated by the numbers in brackets. In **(B)** unpaired two-tailed t-tests and in **(C)** Mann-Whitney tests were used to determine statistical significance. p-values are indicated as * p ≤ 0.05; ** p ≤ 0.01; *** p ≤ 0.001. The scale bar is 5 μm.

in LIN-1(K10A, K169A)::GFP levels was observed after AID::SMO-1 degradation, suggesting that the SUMO pathway may also regulate LIN-1 levels in the 2˚ cells indirectly (**Figs 5A** and **5B** and **S3A and S3B**).

The reduction in wild-type, but not K10A, K169A double mutant LIN-1::GFP expression after AID::SMO-1 degradation suggested that the proteasomal degradation of SUMO can lead to the simultaneous degradation of a SUMO-modified target protein. Since LIN-1(K10A, K169A)::GFP levels were already reduced in the absence of auxin, sumoylation may stabilize LIN-1 in the VPCs. Though, we cannot exclude the possibility that the K10A, K169A mutations may also affect other post-translational modifications of LIN-1, such as acetylation, methylation or ubiquitination, which could affect LIN-1 levels, or that a substitution of one or

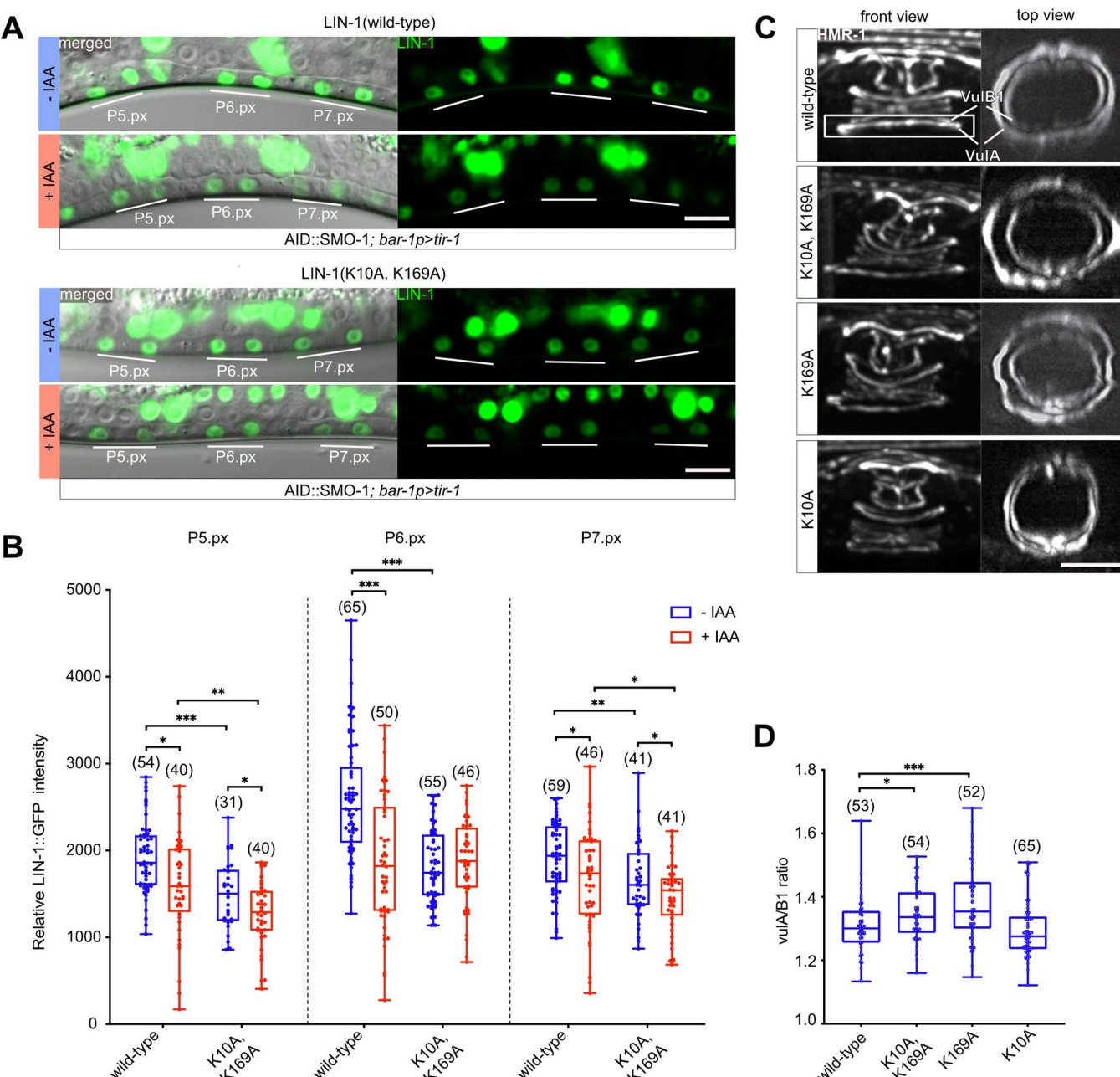

**Fig 5. LIN-1 sumoylation is required for ventral toroid contraction. (A)** Wild-type and K10A, K169A mutant LIN-1::GFP expression in L3 larvae at the Pn. px stage after VPC-specific degradation of AID::SMO-1 from the L2 stage onward. The 1˚ and 2˚ VPC descendants are underlined in white. The left panels show the corresponding DIC images overlaid with the LIN-1::GFP signal in green. **(B)** Quantification of LIN-1::GFP expression levels in 1˚ and 2˚ VPC descendants at the Pn.px stage in LIN-1::GFP wild-type and K10A, K169A double mutants under the indicated conditions. See **S3 Fig** for the corresponding measurements at the Pn.pxx stage. **(C)** Toroid morphogenesis defects in LIN-1 K10A and K169A single and double mutants at the L4 stage. Left panels show lateral views of z-projections. vulA and vulB1 toroids are outlined by the white rectangle in the top left panel and shown in top (xz) views in the right panels. **(D)** Quantification of vulA contraction, calculated as the ratio of the vulA and vulB1 toroid diameter. The box plots show the median values with the 25th and 75th percentiles and the whiskers indicate the maximum and minimum values. Where indicated, untreated controls are labelled with–IAA (blue) and animals treated with 1 mM auxin with +IAA (red). In each graph, the numbers of animals scored are indicated by the numbers in brackets. Statistical significance in **(B)** and **(D)** was calculated with unpaired two-tailed t-tests. p-values are indicated as $^*$ p $\leq$ 0.05; $^{**}$ p $\leq$ 0.01; $^{***}$ p $\leq$ 0.001. The scale bars are 10 μm.

both lysine residues with alanine causes a conformational change in LIN-1 that reduces its stability.

## The LIN-1 K169 SUMO site is necessary for ventral toroid contraction

To investigate the relevance of the two SUMO sites in LIN-1, we introduced the HMR-1::GFP adherens junction marker into *lin-1* single and double SUMO site mutants and investigated toroid formation. (*lin-1(zh157)* and *lin-1(zh158)* refer to the K10A and K169A single SUMO site mutants, respectively, and *lin-1(zh159)* to the double mutant.) In *lin-1(zh159)* K10A, K169A double and *lin-1(zh158)* K169A single mutants, we observed similar toroid contraction defects as seen after AID::SMO-1 or AID::GEI-17 degradation. Specifically, the ratio of the vulA to vulB1 diameter was increased in *lin-1(zh159)* double and *lin-1(zh158)* single mutants (**Fig 5C** and **5D**). By contrast, we did not detect any toroid morphogenesis defects in *lin-1 (zh157)* K10A single mutants.

Thus, only the K169 SUMO site appears to be relevant for a specific aspect of LIN-1 function during vulval toroid morphogenesis. While sumoylation of LIN-1 at K169 may be required for the proper contraction of the ventral vulA toroids, none of the other vulval defects associated with the inhibition of the SUMO pathway were detected in the LIN-1 SUMO site mutants.

## Discussion

### A tissue-specific degradation toolkit to study the SUMO pathway and its targets

Posttranslational protein modification via the SUMO pathway is essential for many biological processes [33]. However, the pleiotropic effects as well as the transient and reversible nature of protein sumoylation have made it difficult to study specific functions of the SUMO pathway. The identification of SUMO targets is usually performed by proteomic approaches [34,35] or through in vitro experiments, but the possibilities to validate candidate SUMO substrates in vivo have so far been limited.

Here, we applied the auxin-inducible protein degradation system AID [19] to inactivate the SUMO pathway in a tissue-specific and temporally controlled manner. This approach may allow the verification of relevant SUMO targets in tissues of interest by following their expression levels after AID::SUMO degradation and observing the resulting phenotypes.

Using the *C. elegans* vulva as a model for organogenesis, we dissected the pleiotropic effects of the SUMO pathway. We induced degradation of endogenously AID-tagged alleles of the SUMO homolog SMO-1 and the SUMO E3 ligase GEI-17 in the different cell types contributing to the vulva. For this purpose, we generated four tissue-specific TIR-1 driver lines to induce AID in the tissues of interest. Tissue-specificity and efficacy of the AID system was confirmed by following the degradation of *gfp*-tagged SUMO E3 ligase GEI-17 at cellular resolution and by quantifying GEI-17 and SMO-1 protein levels via Western blot analysis. The inducible nature of the AID system allowed us to assess the spatial and temporal requirements for protein sumoylation at different stages of vulval development. The stronger penetrance of defects observed after degrading SMO-1 in the VPCs compared to the AC suggested that sumoylation is predominantly required in the VPCs. This observation is consistent with previously reported roles of SMO-1 during vulval development [21,22,36]. Moreover, temporally controlled depletion of SMO-1 indicated that protein sumoylation is continuously required throughout vulval development, controlling a variety of processes like VPC fate specification, AC positioning, BM breaching and vulval toroid morphogenesis. These findings expand the

range of previously reported SUMO phenotypes and provide new insights in the role of sumoylation during vulval development.

Overall, global degradation of either SMO-1 or GEI-17 resulted in stronger phenotypes than AC- or VPC-specific degradation. This suggests that the SUMO pathway has cell non-autonomous functions in other tissues besides the AC and VPCs to control vulval development. For example, neurons in the ventral nerve cord are known to secrete AC guidance cues [37], while adjacent muscles can affect VPC fate specification [38]. Degradation of SMO-1 caused more severe phenotypes than GEI-17 degradation, which could be due to the fact that SMO-1 is the only *C. elegans* SUMO homolog, while GEI-17 is one of several E3 ligases. GEI-17 might be replaced by MMS-21 [39], or sumoylation of certain targets does not require an E3 ligase [10]. Moreover, proteasomal degradation of AID::SMO-1 appears to lead to the simultaneous degradation of sumoylated target proteins, as shown here for the case of LIN-1. This could be another factor explaining the stronger phenotypes observed after SMO-1 degradation compared to GEI-17 depletion.

The identification of SUMO targets is usually performed by proteomic approaches [34,35] or through in vitro experiments. The possibilities to validate candidate SUMO substrates in vivo have so far been limited. The tissue-specific AID approach presented here may allow the verification of relevant SUMO targets in specific cell types by following their expression levels after AID::SMO-1 degradation and observing the resulting phenotypes. Since SUMO targets go through rapid cycles of sumoylation and de-sumoylation, repeated modification with an AID::SMO-1 tag may lead to the progressive proteasomal degradation of a significant proportion of a SUMO target protein, even if only a small fraction is sumoylated at any given time point.

## The SUMO pathway is required for proper BM breaching by the AC

After specification of the VPC fates and before the onset of vulval morphogenesis, the AC breaches two BMs separating the uterus from the vulval cells and invades at the vulval midline in between the 1° vulF cells. Global but neither VPC- nor AC-specific degradation of GEI-17 or SMO-1 resulted in characteristic AC invasion defects, such as the displacement of the AC from the vulval midline and asymmetric BM breaching (**Fig 6**). Occasionally, the AC even failed to breach the underlying BMs. However, we were unable to pin-point a single tissue, in which sumoylation affects AC positioning.

After AC invasion, the VPC continue to proliferate and invaginate, thereby enlarging the breach in the BM. The BMs then slide over the dividing vulF and vulE cells and are stabilized over then un-divided vulD cells, where the INA-1/PAT-3 integrins and the VAB-19 adhesion protein are expressed [30,40]. BM sliding also depends on ventral uterine cells adjacent to the AC. LIN-12 Notch signaling in the uterine π cells upregulates *ctg-1* expression, which allows BM sliding by downregulating the dystroglycan BM-adhesion receptor [41]. As reported by Broday et al. [11] and consistent with our observations, sumoylation is required for the formation of a uterine lumen. The abnormal connection between the vulva and uterus could in part be caused by a loss of sumoylation of LIN-11 in the π cells. AC positioning, on the other hand, depends on guidance signals from both the VPCs and the ventral nerve cord that polarize the AC along the dorso-ventral axis [37,42]. We thus propose that the mispositioning of the AC and asymmetrical BM breaching are caused by a combination of defects in multiple tissues.

## The SUMO pathway in the VPCs controls vulval toroid morphogenesis

The most penetrant class of phenotypes caused by disruption of the SUMO pathway affects vulval toroid morphogenesis. All toroid morphogenesis defects could be observed with the

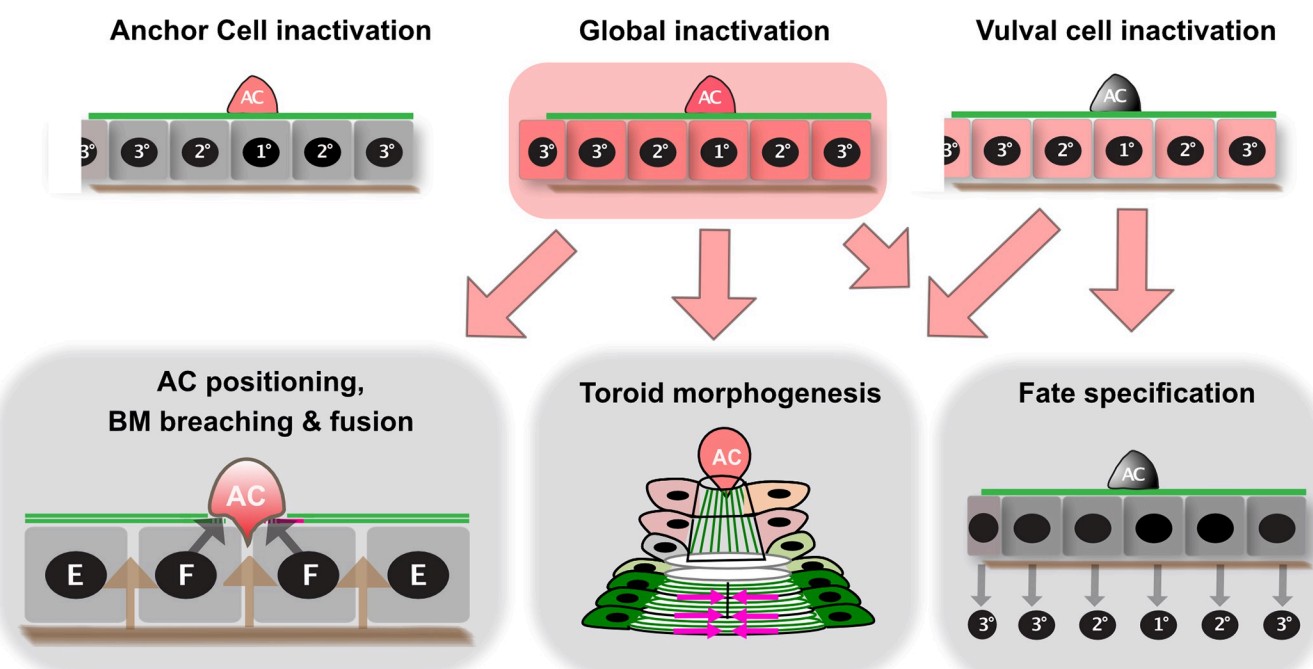

**Fig 6. Overview of the different phenotypes observed after SUMO pathway inactivation.** Global inactivation of the SUMO pathway (center top) perturbs AC positioning, BM breaching and AC fusion in a cell non-autonomous manner, as well as vulval morphogenesis and fate specification. The magenta arrows indicate the direction of contractile forces shaping the toroids. Vulva-specific SUMO inhibition (top right) results in abnormal morphogenesis of the ventral toroids and rare vulval fate specification defects. Inhibition of SUMO only in the AC (top left) does not cause distinctive vulval defects.

*bar-1p>tir-1* driver, indicating that these phenotypes are likely due to a cell-autonomous function of the SUMO pathway in the vulval cells (**Fig 6**). Inhibiting the SUMO pathway altered *egl-17* gene expression in the dividing 1˚ VPCs already before the morphogenesis phase, indicating an involvement of SUMO pathway during VPC fate specification [20,21,36]. Moreover, we observed rare defects in proximal VPC induction and an ectopic induction of the posterior VPC P8.p after inhibition of the SUMO pathway, which also points to a role in VPC fate specification. Even though we could not directly correlate VPC induction with AC positioning, it is possible that the ectopic VPC induction is at least in part due to the AC mispositioning.

The observed vulval morphogenesis defects were almost fully penetrant, indicating that the SUMO pathway is most relevant during morphogenesis, after the VPC fates have been specified. Protein sumoylation is required for different aspects of vulval morphogenesis, such as the formation of the correct connections and fusion between the contralateral pairs of vulval cells and for the contraction of the ventral vulA toroids (**Fig 6**).

## The LIN-1 sumoylation site K169 is necessary for the contraction of the ventral vulval toroids

The ETS family transcription factor LIN-1 is a well-characterized SUMO target, originally identified in genetic screens for mutants with abnormal vulval development [20–22]. While a complete loss of *lin-1* function causes a completely penetrant Muv phenotype due to loss of its repressor function, *lin-1(lf)* mutations also cause reduced *egl-17* reporter expression in the 1˚ VPC lineage [43]. Moreover, LIN-1 promotes ventral toroid contraction by inducing expression of the RHO kinase LET-502 in the 2˚ toroids [23]. Together, these findings suggested that loss of LIN-1 sumoylation could be responsible for a subset of the defects caused by inhibition

of the SUMO pathway. Consistent with this hypothesis, we observed reduced expression levels of wild-type LIN-1 in the 1˚ vulval cells after degradation of AID::SMO-1, while mutation of the sumoylation sites K10 and K169 rendered LIN-1 insensitive to AID::SMO-1 degradation. Even though direct biochemical evidence demonstrating sumoylation of LIN-1 in vivo is lacking, mainly due to the very low endogenous LIN-1 expression levels, our data suggest that LIN-1 is indeed sumoylated in the 1˚ vulval cells. Moreover, mutation of the K169 sumoylation site in LIN-1 caused similar defects in vulA toroid contraction as VPC-specific inhibition of the SUMO pathway. We thus propose that sumoylation of LIN-1 at K169 is necessary for this specific activity during vulval toroid formation. Vulval fate specification, on the other hand, was not visibly affected by mutation of either of the two SUMO sites in LIN-1.

The very specific phenotype of LIN-1 sumoylation site mutants suggests that the pleiotropic effects on vulval morphogenesis caused by perturbation of the SUMO pathway can be attributed to the combined action of a number of distinct SUMO targets acting in different tissues.

Although *C. elegans* LIN-1 and other ETS family transcription factors share only limited sequence homology, several mammalian ETS transcription factors including Ets-1 and Elk-3/Net were found to be sumoylated at various sites [44–47]. It thus appears that SUMO modification of the LIN-1/ETS family of transcription factors is a conserved process in metazoans, regulating their transcriptional activity in conjunction with phosphorylation.

## Material and methods

### *C. elegans* handling and maintenance

*Caenorhabditis elegans* strains were grown on standard NGM (Nematode Growth Medium) plates seeded with OP50 *E. coli* bacteria and incubated at 15˚C, 20˚C or 25˚C as indicated [48]. The derivate of Bristol strain N2 was used as a wild-type reference. A list of strains used in this study is provided in **S1 Table**.

### Design of the tissue-specific SUMO degradation toolkit

All TIR-1 degradation drivers were generated with an analogous design in the pCFJ151 backbone and integrated by MosSCI in selected genetic locations. To track the tissue-specificity of each construct, we used an SL2 trans-splicing domain followed by an mCherry reporter (fragment derived from pSA120 [49]) to express the fluorophore under the same promoter as TIR-1. In all constructs, we used the *unc-54* 3'UTR. TIR-1 was amplified from pLZ31 [19]. The following promoters/enhancers were used: the *egl-17* promoter was amplified as a 2042 bp fragment from a derivate of pPD107.94/mk84-148 [50], the *cdh-3* promoter was amplified as a 1897 bp fragment from a derivate of pPD104.97/mk62-63 [37], the *bar-1* promoter as 3216 bp fragment [51] and the *hlh-2prox* promoter as a 576 bp fragment driving the expression in two alpha and two beta cells (3VU and 1AC) [24]. The promoters/enhancers are indicted in text as *egl-17p*, *cdh-3p*, *bar-1p* and *hlh-2p*. All constructs were cloned by Gibson assembly [52]. The following plasmid constructs were microinjected at the indicated final concentrations into young adult EG6699, EG8078 or EG8080 hermaphrodites: transgene in pCFJ150: 50 ng/μl, transformation markers pGH8 (*rap-3p>mCherry*): 10 ng/μl, pCFJ104 (*myop-3>mCherry*): 5 ng/μl, pCFJ90 (*myo-2p>mCherry*): 2.5 ng/μl and pJL43.1 expressing Mos1 transposase: 50 ng/μl [53]. The transformants were screened for crawling animals, which lacked the co-injected transformation markers and genotyped for homozygous insertion by PCR. The list of plasmids generated and primers used for amplification of selected fragments and genotyping can be found in **S2 and S3 Tables** in the **Supporting information**.

## CRISPR/Cas9 genome editing

For CRISPR/Cas9 editing, the protocol by Dickinson et al. [54] was followed. Plasmids containing the repair template and single guide RNAs were used at a concentration of 10 ng/μl and 50 ng/μl, respectively. We used the same transformation markers at the same concentrations as for MosSCI insertions. To generate the *smo-1(zh140)* allele, an oligonucleotide corresponding to a target sequence near the *smo-1* translational start site (sgRNA: GCC GAT GAT GCA GCT CAA GC) was cloned into the plasmid pMW46 (derivate of pDD162 from Addgene). The 5'homology arm was amplified from genomic DNA with OAF239 and OAF344. The 3'homology arm was amplified with OAF345 and OAF346. The AID sequence was cloned from pLZ29 with OAF334 and OAF335 to insert the 45 amino acid Degron sequence and 3xFLAG tag at the SMO-1 N-terminus. The backbone of plasmid containing the Self-Excising Selection Cassette was amplified in two fragments with OAF339/ OAF340 and OAF343/ OAF337 from pDD282.

To generate the *gei-17(zh142)* allele, an oligonucleotide corresponding to a target sequence near the *gei-17* translational start site (sgRNA: GTC GTT TCG AGA CAC AGC GG) was cloned into the plasmid pMW46. The 5'homology arm was amplified from genomic DNA with OAF336 and OAF338. The 3'homology arm was amplified with OAF341/ OAF342. The backbone containing the Self-Excising Selection Cassette and AID sequence was cloned in two fragments with OAF334/ OAF340 and OAF343 /OAF337 from pAF56, a previously cloned repair template for AID::SMO-1.

To generate the LIN-1 sumoylation site mutants *lin-1(zh157)* (K10A), *lin-1(158)* (K169A) and *lin-1(zh159)* (K10A, K169A), genome editing was performed according to the co-CRISPR strategy described by Arribere et al. [55]. To introduce the K10A mutation an oligonucleotide corresponding to a target sequence (sgRNA: GTC GAG TTC GGA AGA AGC CG) was cloned into plasmid pMW46. To introduce K169A mutation an oligonucleotide corresponding to a target sequence (sgRNA: GTT CAT ATT TGA GGA AAA GT) was cloned into the plasmid pMW46. The following constructs with indicated final concentration were microinjected into young adult *lin-1(st12212)* hermaphrodites: *dpy-10* sgRNA pJA58 (25 ng/μl), *dpy-10* repair oligonucleotide AF-ZF-827 (0.5 nM), *lin-1* sgRNA (75 ng/μl), *lin-1* repair oligonucleotide OAF377 (0.5 nM, introducing an NruI restriction site for K10A) or OAF378 (0.5 nM, introducing a SacII restriction site for K169A). To generate the *zh159* double mutant, the sgRNA#4 plasmid and OAF378 repair oligonucleotide was injected with the *dpy-10* sgRNA plasmid and *dpy-10* repair oligonucleotide into *lin-1(157)* hermaphrodites at the same concentrations as for the single mutant. Transformants showing a Rol phenotype were transferred to separate NGM plates, and animals containing the desired point mutations were identified by PCR amplification using the primers OAF365/ OAF366 for K10A or OAF367/ OAF368 for K169A, followed by restriction digests with NruI or SacII, respectively. The new *lin-1* alleles were sequenced and back-crossed three times to N2.

## Auxin treatment

NGM plates containing 1 mM auxin were prepared according to Zhang et al. [19], seeded with OP50 *E. coli* bacteria and used immediately for the experiments. The auxin treatment protocol was adapted for each strain due to the differences in strain viablility and fertility. For strains containing AID-tagged GEI-17 (*gei-17(fgp1)* and *gei-17(zh142)* alleles), animals were synchronized by bleaching, and hatched L1 larvae were plated on auxin or control plates. Control plates contained the same dilution of ethanol, in which the auxin stock solution was prepared, as auxin plates. Animals were incubated at 25˚C and analyzed after 24 h or 36 h of treatment during the L3 or adult stage, respectively. Since homozygous *smo-1(zh140)* animals are sterile,

they were maintained balanced with *tmC20*, and homozygous *smo-1(zh140)* animals were selected for the experiments. AID-tagged SMO-1, animals were likewise synchronized by bleaching, but hatched L1 larvae were first plated on standard NGM plates containing OP50 and incubated at 20°C for 24 h, followed by transfer to auxin or control plates and 24 h of treatment. L3 animals were imaged right after treatment was complete, animals analyzed in the adult stage were instead transferred to standard NGM plates and analyzed 24 h later. For experiments involving different treatment periods, *smo-1(zh140)* animals were put on auxin/control plates 12 h, 24 h, 30 h and 36 h after L1, and transferred back to standard NGM plates 48 h after L1. Homozygous adults were analyzed 24 h later.

## Western blot analysis of auxin-induced protein degradation

Fourty adult animals were transferred to an Eppendorf tube containing 20 μl of water. 20 μl of 2xSDS buffer were added, and the sample was incubated for 5 min at 95°C. Genomic DNA was digested by adding 1 μl of DNase (Qiagen) and incubating for 5 min at room temperature, followed by a 5 min inactivation at 95°C. Proteins were separated by SDS PAGE on 4–12% acrylamide gradient gels and blotted onto PVDF membranes. After blocking non-specific binding sites with 5% milk or bovine serum albumin in TBST (20 mM Tris, 150mM NaCl, 0.1% Tween 20), the membranes were incubated with the primary antibody diluted in TBST containing 5% milk overnight at 4°C. After incubation with HRP-conjugated secondary antibodies, the protein bands were viualized by chemiluminescence using the SuperSignal West Pico or Dura Chemiluminescent Substrate (Thermo Scintific). Quantification was performed by measuring the band intensities using Fiji's measurement tools [56]. Band intensities were first normalized to the alpha-tubulin loading controls and then to the maximum value in each experiment. The following antibodies were used: anti-SUMO-1 1:500 (S5446 Sigma), anti-Flag 1:3000 (Sigma F3165-1MG), anti-Tubulin 1:10 000 (Abcam ab18251), HRPGoat anti-Rabbit 1:2000 (Jackson ImmunoReserach 111-035-144) and HRP Goat anti-Mouse 1:2000 (Jackson ImmunoReserach 115-035-146).

## Microscopy and image processing

For Nomarski and fluorescence imaging, live animals were mounted on 4% agarose pads and immobilized with 20 mM tetramisole hydrochloride solution in M9 buffer, unless stated otherwise. For toroid analysis in *lin-1* mutants, we used custom microfluidic devices to immobilize the animals and performed imaging as described [57]. Images were acquired with a Leica DM6000B microscope equipped with Nomarski and fluorescence optics, as well as a Leica DFC360FX camera and 63x (N.A. 1.32) oil immersion lens; a Leica DMRA microscope controlled by a custom build Matlab script, equipped with an image splitter and two Hamamatsu ORCA-flash 4.0LT+ cameras to simultaneously acquire z-stacks in the DIC, mCherry and GFP channels using a 63x (N.A. 1.32) oil immersion lens; or a Matlab controlled Olympus BX61 microscope equipped with a X-light V2 spinning disc confocal system, a Prizmatix UHP-T-460-DI/UHP-T-560-DI LED as light source, an Andor iXon ultra888 EMCCD camera and a 60x (N.A 1.3) or 100x Plan Apo (N.A 1.4) oil immersion lens. Images were analyzed and quantified with Fiji software [56].

## Scoring vulval induction and morphogenesis

The numbers of induced VPCs was scored in synchronized L4 animals as described in Schmid et al. [58]. A score of 1 was assigned to a VPC when it underwent three division rounds and 0.5 when only one of the two VPC descendants had differentiated. A score of 0 was assigned to uninduced VPCs that had divided once and fused with the hypodermis.

Vulval lumen morphogenesis was assessed based on DIC microscopy at the L4 stage (L4.3-L4.7). Vulval defects in adult animals were scored by using a dissecting scope. Any abnormality in the vulval tissue visible under a dissecting microscope was categorized as 'abnormal vulva'phenotype.

## Analysis of toroid formation

Animals at the L4 stage were imaged either on agar pads or in microfluidic devices [59] at 60x or 100x magnification, and z-stacks with a spacing of 0.13 to 0.2 μm were acquired. Toroid formation was monitored using the *swIs79[ajm-1::gfp]* or *cp21[hmr-1::gfp]* adherens junction markers [31,32]. Images were deconvolved either by the Huygens deconvolution software (Scientific Volume imaging) or using the Deconvolution lab plugin in Fiji [56]. The measurement of the vulA and vulB1 diameters was done in xz-views of the cropped ventral toroids. The toroid fusion defects were scored in 3D reconstructed z-stacks.

## Quantification of LIN-1::GFP and EGL-17::YFP expression levels

Animals at the mid-L3 stage were imaged at 63x magnification using a wide-field microscope, acquiring z-stacks with a spacing of 0.3 μm. The average intensity of the nuclear LIN-1::GFP signal was measured in background-subtracted, summed z-projections of 3 mid-sagittal sections of the VPCs. The nuclei of the 1˚ and 2˚ VPC descendants were manually selected, and the mean nuclear signal intensity was measured using the built-in measurement tools in Fiji. The data represent the averaged measurements for each VPC lineage (two nuclei at the Pn.px and four at the Pn.pxx stage). *egl-17::yfp* expression levels were analyzed in background subtracted mid-sagittal sections of the P6.x-P6.xxx cells. Cell bodies were manually selected, and the mean intensity was measured in Fiji.

## AC mispositioning and BM breaching shift analysis

Worms between L4.0-L4.5 were imaged at 63x or 100x magnification and z-stacks with a spacing of 0.1–0.3 μm were acquired. The AC position was monitored based on the *qyIs50[cdh-3>mCherry::moeABD; unc-119(+)]* reporter and DIC images, and the BM breach with the *qyIs10[lam-1>lam-1::gfp]* reporter. To assess the alignment of the AC with the 1˚ VPCs, the angle between a line through the middle of the vulval invagination and the center of the ACs nucleus and the dorso-ventral axis was measured. To quantify the BM breaching shift, the angles α and β between the middle of the vulval invagination to each of the BM breach points were measured and the ratios of the two angles was calculated (as illustrated in Fig 3D).

## Statistical analysis

Statistical analysis was performed using GraphPad Prism as indicated in the figure legends. Data were tested for parametric distribution and outliers were removed from analysis. For non-parametric continuous data, we used the Kolmogorov-Smirnov test, for non-continuous data (e.g. VPC induction counts) the Mann-Whitney test. Numerical values used for statistical analysis can be found in the **S1 Data** Excel file.

## Supporting information

**S1 Fig. The SUMO pathway acts throughout vulval development.**
(TIFF)

**S2 Fig. VPC- and AC-specific degradation of SMO-1 do not affect AC positioning or BM breaching.**
(TIFF)

**S3 Fig. LIN-1:GFP expression is regulated by the SUMO pathway at the Pn.pxx stage.**
(TIFF)

**S1 Table. Strains used.**
(DOCX)

**S2 Table. Plasmids used with details on the plasmid constructions.**
(DOCX)

**S3 Table. List of primers used.**
(DOCX)

**S4 Table. Number of animals scored in three independent replicates for S1A Fig.**
(DOCX)

**S5 Table. Number of animals scored in three independent replicates for S1B Fig.**
(DOCX)

**S1 Data. Numerical data used for the figure graphs and statistical analysis.**
(XLSX)

## Acknowledgments

We would like to thank all members of the Hajnal laboratory, Frauke Melchior, Damian Brunner and Ulrike Kutay for input, and the *Caenorhabditis Genetics Center* (funded by NIH Office of Research Infrastructure Programs (P40 OD010440)) for providing strains.

## Author Contributions

**Conceptualization:** Aleksandra Fergin, Alex Hajnal.

**Data curation:** Aleksandra Fergin.

**Formal analysis:** Aleksandra Fergin.

**Funding acquisition:** Alex Hajnal.

**Investigation:** Aleksandra Fergin, Gabriel Boesch, Nadja R. Greter, Alex Hajnal.

**Methodology:** Aleksandra Fergin, Gabriel Boesch.

**Project administration:** Alex Hajnal.

**Supervision:** Alex Hajnal.

**Visualization:** Aleksandra Fergin.

**Writing – original draft:** Aleksandra Fergin.

**Writing – review & editing:** Simon Berger, Alex Hajnal.

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
