## [Decision Letter · Decision Letter 0]

3 Feb 2022

Dear Dr Hajnal,

Thank you very much for submitting your Research Article entitled 'Tissue-specific inhibition of protein sumoylation uncovers diverse SUMO functions during C. elegans vulval development' to PLOS Genetics.

The manuscript was fully evaluated at the editorial level and by four independent peer reviewers. The reviewers appreciated the attention to an important topic but identified some concerns that we ask you address in a revised manuscript. In particular, note that reviewer #1 strongly suggests to provide biochemical evidence showing that using the AID system you can directly detect a change in the SUMOylation state of at least one target protein.

We therefore ask you to modify the manuscript according to the review recommendations. Your revisions should address the specific points made by each reviewer.

[LINK]

Yours sincerely,

Benjamin Podbilewicz

Guest Editor

PLOS Genetics

Gregory P. Copenhaver

Editor-in-Chief

PLOS Genetics

Reviewer's Responses to Questions

**Comments to the Authors:**

Reviewer #1: In this manuscript, Fergin et al describe a system to genetically dissect, in a spatiotemporal fashion, the role of essential proteins. The system is based on the degradation of proteins of interest by the auxin-inducible protein degradation system (AID). The authors choose to study the role of SUMO in the development of C. elegans vulva during the postembryonic stages. The AID system opens a new avenue for the study of many important proteins that are essential for organism viability and have multiple roles that hinder classical genetic dissection using a traditional loss of function mutations. In this context, the decision to focus on SUMO and C. elegans vulva development serves well the proof of principle aim of this paper. Essential for C. elegans reproduction, SUMO acts multiple times during C. elegans development affecting many biological processes. C. elegans vulva is probably the most studied organ of all animals; thus, it was and still is a fruitful ground for new discoveries. Using this approach, Fergin et al identify some of the processes (i.e. basement membrane breaching) that are dependent on the activity of SUMO along with some of its targets (i.e. LIN-1).

Major comments:

1) I am not convinced that the manuscript is within the exact scope of PLOS Genetics. The novelty of this work is primarily in applying the AID system for the study of the role of pleiotropic proteins in vivo. I think that a journal focusing on techniques and methods will be more suitable for this type of paper.

2) There is no biochemical evidence for the activity, effectiveness, and specificity of the system. One major gold standard of the SUMO field is to show biochemically that a protein claimed to be SUMOylated is indeed SUMOylated in vivo. In this manuscript, there is no biochemical evidence want so ever for the activity of the system. Morphological analysis can hint at the process affected but provides only the end result of the effect. Measuring the GFP levels of target proteins does not demonstrate directly that these proteins are SUMOylated in wild-type conditions and not SUMOylated when the AID system is active. The cehnges in the levels/ position of the GFP signal may stem from the activity of SUMO on a regulator of the proposed target gene. I would strongly recommend showing biochemically that the system can change the SUMOlyation state of at least one target protein.

3) The AID system is a well-established system used in C. elegans for a couple of years now for spatiotemporal knockdown of protein of interest. Moreover, the role of SUMO in the development of C. elegans vulva is quite well-characterized starting from 2004. I am not convinced that the manuscript really adds enough new data to justify a publication in PLOS genetics. The novelty of this manuscript is in combining the AID system with careful morphological and expression studies but the biological significance of the manuscripts remains in the realm of a proof of principle.

Reviewer #2: In this study the authors used the auxin-inducible protein degradation system to inhibit sumoylation either in the C. elegans anchor cell (AC) or the vulva. Four different promoters regulating the TIR-1 ubiquitin E3 ligases were used to inhibit the SUMO system before and after AC specification, in the primary VPCs and in all VPCs. In addition, a somatic promoter was used to degrade SUMO in somatic tissues. This study proved that the TIR-1 system is working in the vulva, but new findings are limited as the specific drivers did not resulted in phenotypes therefore could not dissect the possible different functions and targets of SUMO in P6.p and in P5.p and P7.p and also the degradation was not enough or is not required in the AC.

Comments:

1. Figure 1B: the insets could be bigger as it is difficult to see the degradation of the GFP happening with the small insets. Also, the insets could be shown for every promoter rather than just the first promoter.

2. Figure 2C. The data on AID::SUMO is not shown.

3. Figure 2C. SUMO degradation cause ectopic posterior vulva while GEI-17 did not, did the authors suspect specific activity of another sumo ligase?

4. LIN-1 - what was the reasoning to change K to alanine residues and not to arginine? Maybe LIN-1 was destabilized as the charge was changed and another option is that the auxin system could be leaky specifically on this target.

5. There is no evidence in the data presented that substantial fraction of endogenous LIN-1 is sumoylated in the 1ᵒ VPC and that degradation of SUMO lead to degradation of its substrates.

Additional comments:

Line 50- sumo is attached to substrates also without E3 ligase (in contrast to ubiquitin)

Line 62- comment typo

Line 82- it is hard to hypothesize this as sumylation is transient and only a small fraction of a substrate is sumoylated. This system is expected to degrade free SUMO, this will cause a decrease in sumoylation of targets.

Reviewer #3: The manuscript by the Hajnal group describes a profound analysis of the influence of sumoylation on C. elegans vulva development. The authors use the most sophisticated technology to create tissue-specific constructs to manipulate sumoylation processes. Their work provides an important increase in our understanding of the complex gene regulatory network controlling vulva development.

In general, C. elegans vulva development represents one of the best-studied developmental processes in all of animal development. Originally chosen for its simplicity the vast amount of data collected during the 1980ies and 2010ies has resulted in an extremely detailed understanding of i) cellular interactions, ii) the involvement of signaling pathways, iii) feedback loops, and finally, post-translational modifications. This detailed work has highlighted the importance of two general genetic principles, pleiotropy and redundancy, which, given its consequential complexity has resulted in many labs leaving the field. The Hajnal group is an important exception that is taking the challenging efforts to further our understanding of various aspects of vulva development.

This manuscript focuses on the role of sumoylation as one post-translational regulatory process. Using tissue and cell-specific constructs employing the auxin-inducible protein degradation system, the authors can show multiple roles of sumoylation in a sophisticated temporal and spatial dimension. Confirming the complexity of the regulation of the vulva-GRN, the results of the study are not always super clean because there are simply too many molecular processes requiring this type of post-translational modification.

This work was carried out extremely careful. The first paragraphs of the result section establish the constructs to be used and provide a state-of-the-art methodology. This analysis alone is worth its independent publication. What follows is a careful analysis that is properly performed and discussed without any overstatements. I have nothing to criticize on the experimental design, the data representation and careful discussion.

Given the broad scope of this manuscript, I have only two recommendations to the authors. Given that vulva papers are rare these days, the manuscript would profit from two philosophical additions in the introduction and the discussion:

1. In the introduction, I would urge the authors to add a more detailed description of pleiotropy and redundancy as general principles in genetics. Vulva development in C. elegans provides some of the best examples in all of animal development (besides Drosophila). I find it an important point to highlight that what the authors call ‘complexity’ is in reality, in large parts, the combination of pleiotropy and redundancy of several genes and signaling pathways involved in the regulation of vulva development. As this manuscript will likely be the most modern on vulva development for some time, this addition might be important.

2. In the Discussion, the manuscript would profit from the addition of an evolutionary interpretation of the authors findings; both, with regard to the evolution of vulva development and the evolution of the role of sumoylation in animal development.

In summary, this is an exceptional contribution, very careful designed, performed and interpreted. I highly recommend publication in PLoS Genetics.

Ralf J. Sommer

Reviewer #4: A major challenge in understanding the function of broadly expressed, broadly active genes (such as those that encode core cellular functions) in development is to independently alter their activity. This study clearly shows why this approach is useful and how appropriate tools can help dissect such complexity. Experimentally, I have no issues. I like the LIN-1 experiments!

Major comments

1. I would like to see a concise summary of the tissues/cells examined and the effects to make it easier for the non-vulval specialist to comprehend how the pieces fit together. It might be good as a table with cell type, phenotype, and whether affected.

2. Some effects might be independent or downstream knock-on effects of a primary perturbation especially since there is extensive cross-talk among the targeted cell types/stages. Specifically, what is the minimum set of independent effects you see and what might be the maximum. Are all phenotypes autonomous (based on existing knowledge in the field) This can be set in the context of the summary I requested in point 1.

minor comment

The specific choice of references seems odd. I think they are overall fine, but I just wanted to make sure the choices of which specific primary articles and reviews to cite were conscious and not random.

**Have all data underlying the figures and results presented in the manuscript been provided?**

Reviewer #1: Yes

Reviewer #2: Yes

Reviewer #3: Yes

Reviewer #4: Yes

PLOS authors have the option to publish the peer review history of their article (what does this mean?). If published, this will include your full peer review and any attached files.

Reviewer #1: No

Reviewer #2: No

Reviewer #3: **Yes: **Ralf J. Sommer

Reviewer #4: No

---

## [Editor Report · Decision Letter 1]

13 May 2022

Dear Dr Hajnal,

We are pleased to inform you that your manuscript entitled "Tissue-specific inhibition of protein sumoylation uncovers diverse SUMO functions during C. elegans vulval development" has been editorially accepted for publication in PLOS Genetics. Congratulations!

Yours sincerely,

Benjamin Podbilewicz

Guest Editor

PLOS Genetics

Gregory P. Copenhaver

Editor-in-Chief

PLOS Genetics

Comments from the reviewers (if applicable):

**Data Deposition**

http://datadryad.org/submit?journalID=pgenetics&manu=PGENETICS-D-21-01598R1

**Press Queries**

---

## [Editor Report · Acceptance letter]

1 Jun 2022

PGENETICS-D-21-01598R1 

Tissue-specific inhibition of protein sumoylation uncovers diverse SUMO functions during C. elegans vulval development 

Dear Dr Hajnal, 

We are pleased to inform you that your manuscript entitled "Tissue-specific inhibition of protein sumoylation uncovers diverse SUMO functions during C. elegans vulval development" has been formally accepted for publication in PLOS Genetics! Your manuscript is now with our production department and you will be notified of the publication date in due course.

With kind regards,

Livia Horvath

PLOS Genetics

On behalf of:
